# Validation of an Ultra-Wideband Tracking System for Recording Individual Levels of Activity in Broilers

**DOI:** 10.3390/ani9080580

**Published:** 2019-08-20

**Authors:** Malou van der Sluis, Britt de Klerk, Esther D. Ellen, Yvette de Haas, Thijme Hijink, T. Bas Rodenburg

**Affiliations:** 1Animal Breeding and Genomics, Wageningen University & Research, 6700 AH Wageningen, The Netherlands; 2Department of Animals in Science and Society, Faculty of Veterinary Medicine, Utrecht University, 3508 TD Utrecht, The Netherlands; 3Research &Development Department, Cobb Europe BV, 5831 GH Boxmeer, The Netherlands; 4Adaptation Physiology Group, Wageningen University & Research, 6700 AH Wageningen, The Netherlands

**Keywords:** tracking, broilers, activity, ultra-wideband, group housing

## Abstract

**Simple Summary:**

Broiler chickens are often kept in large groups, which makes it difficult to identify individual birds and monitor their activity. Here, we studied whether an automated tracking system, using ultra-wideband technology, could be implemented to study activity of individual broilers. We compared the distance as recorded with the tracking system to the distance recorded on video and found a moderately strong positive correlation. Using the tracking system, we were able to detect decreases in activity over time, and we found that lightweight birds were on average more active than heavier birds. Both these results match with reports from literature and therefore support the conclusion that the tracking system appears well-suited for monitoring activity in broilers. The information on activity over time that can be collected with this system can potentially be used to study health, welfare and performance at the individual level, but further research into individual patterns in activity is required.

**Abstract:**

Individual data on activity of broilers is valuable, as activity may serve as a proxy for multiple health, welfare and performance indicators. However, broilers are often kept in large groups, which makes it difficult to identify and monitor them individually. Sensor technologies might offer solutions. Here, an ultra-wideband (UWB) tracking system was implemented with the goal of validating this system for individual tracking of activity of group-housed broilers. The implemented approaches were (1) a comparison of distances moved as recorded by the UWB system and on video and (2) a study recording individual levels of activity of broilers and assessing group-level trends in activity over time; that could be compared to activity trends from literature. There was a moderately strong positive correlation between the UWB system and video tracking. Using the UWB system, we detected reductions in activity over time and we found that lightweight birds were on average more active than heavier birds. Both findings match with reports in literature. Overall, the UWB system appears well-suited for activity monitoring in broilers, when the settings are kept the same for all individuals. The longitudinal information on differences in activity can potentially be used as proxy for health, welfare and performance; but further research into individual patterns in activity is required.

## 1. Introduction

In current husbandry systems, animals are often kept in large groups. This makes it difficult to identify and monitor animals at the individual level; especially when animals in a group are homogeneous in appearance, and automatic visual identification of animals is hampered. Often, video recordings are used to manually assess animal behaviour, but the manual analysis of these videos is time-consuming and may introduce human error [1].

Even though monitoring of individuals is difficult, there is a growing interest in quantifying individual behaviours of group-housed animals, in order to study the link between individual behaviour and performance in more detail. Broiler chickens are an example of a livestock species for which data on individual behaviour could be valuable. A specific trait of interest in broilers is activity. Changes in activity in broilers appear to have potential as a proxy for multiple health, welfare and performance indicators. For example, ill animals may show a change in activity, as they generally increase their time spent sleeping [2]. Furthermore, it has been suggested that low activity levels, as well as higher body weights, are related to lameness or leg weakness [3,4,5]. Leg weakness is marked by an impaired walking ability, or abnormal gait, with a noninfectious cause and may negatively affect the welfare of the birds [6]. However, it has been indicated that increased locomotor activity can contribute to a lower prevalence or severity of leg weakness [7]. There might be potential to select for activity. For example, Bizeray et al. (2000) found differences in early life activity between two broiler stocks and noted that these differences were possibly due to genetic factors [8]. Overall, individual levels of activity appear to be a key trait to monitor. However, as mentioned before, identifying animals and monitoring their behaviour in large groups is difficult.

Sensor technology systems, such as passive radio frequency identification (RFID), automated computer vision (CV) or ultra-wideband (UWB) tracking, might offer solutions for tracking individual animals that are housed in groups. These technologies have already been implemented in several studies focusing on poultry behaviour. In studies involving passive RFID, birds are generally fitted with tags, while antennas are placed in the location of interest to register presence of birds. RFID has been implemented for example to study range use (i.e., pop hole use) in laying hens (e.g., the work by the authors of [9]), and to register nesting and feeding behaviour (e.g., the work by the authors of [10]) or presence in a preference chamber (e.g., the work by the authors of [11]). In studies involving automated CV systems, a camera is often placed above a group to monitor behaviour. For example, Aydin (2017) used a 3D vision camera system with a depth sensor to study inactivity of broilers, by analysing the number of lying events and the latency to lie down of individual broilers walking through a test corridor [12]. In studies involving UWB systems, animals are generally equipped with transmitters, while receivers, which receive the signals from the transmitters, are placed in the surroundings. Stadig et al. (2018) implemented an UWB system to monitor free-range use of chickens, and observed in validation trials that the median error in the localisation of tags was 0.29 m [13]. Rodenburg et al. (2017) compared UWB and automated video tracking to track individual laying hens and found that the results of both were very similar and that the UWB system could detect the bird’s location with an accuracy of 85% [14]. 

Overall, it appears that these technologies all have potential for individual tracking (for a more elaborate review of these sensor technologies and their applicability for poultry, see the work by the authors of [15]). However, to obtain information on activity levels in group-housed broilers, an UWB system appears most practical, as this allows for exact positioning of the animals and provides reliable identification of individuals. Furthermore, the UWB signals are noted to be relatively insensitive to reflections from the surroundings and do not require a direct line of sight [16,17]. This makes the implementation of an UWB system a promising approach for identifying and locating individual group-housed broilers. Furthermore, using the longitudinal information on location that is provided by an UWB system, activity can potentially be assessed. 

In the present study, an UWB tracking system was implemented to track individual activity in broilers, with the goal of validating the UWB system for individual tracking of activity of group-housed broilers. The two approaches implemented to validate the UWB system were (1) a comparison of the distances moved as recorded by the UWB system and the distances recorded on video and (2) a study recording individual levels of activity of broilers and assessing group-level trends in activity over time, which could be compared to known activity trends from literature. Literature on broiler activity indicates that age and weight affect activity in a general direction. Activity decreases are seen with increasing age of broilers [18,19]. For example, Tickle et al. (2018) found changes over the growth period in the proportion of time spent actively moving and in walking speed, both of which declined with increasing age [19]. Furthermore, the weight of birds is thought to affect the level of activity, with a lower level of activity in heavier birds. For example, Reiter & Bessei (2001) studied locomotor activity of 25% load-reduced and non-load-reduced birds and found that the average distance travelled by broilers was higher in load-reduced birds compared to non-load-reduced birds [20]. Therefore, if the UWB system performs well, it would be expected that a decrease in activity over time and a lower level of activity in heavier birds are observed in this study, and that there is a strong correlation between distances recorded with the UWB system and on video.

## 2. Materials and Methods 

### 2.1. Ethical Statement

Data were collected under control of Cobb Europe. Cobb Europe complies with the Dutch law on animal wellbeing. This study is not considered to be an animal experiment under the Law on Animal Experiments, as confirmed by the local Animal Welfare Body (20th of June, 2018, Lelystad, The Netherlands).

### 2.2. Location and Housing

All trials were performed on a broiler farm in The Netherlands. The broilers were housed in groups and feed and water were provided ad libitum. Wood shavings were provided as bedding. The birds were kept under a commercial lighting and temperature schedule, and were vaccinated according to common practice [21].

### 2.3. Ultra-Wideband System

To track individual broilers and to determine their activity levels, an UWB system was implemented. Generally, UWB systems consist of transmitters, which can be fixed to the animal of interest, and receivers in the environment. UWB systems communicate between these transmitters and receivers using narrow radio frequency pulses [16] that are spread out over a spectrum of at least a 500 MHz bandwidth, or have a fractional bandwidth equal to or larger than 20% of the centre carrier frequency [22]. These UWB signals allow for calculating the position of the transmitters using triangulation of the signal between the receivers. Here, a Ubisense UWB system with Series 7000 sensors and compact tags (Ubisense Limited, Cambridge, UK) was used, in combination with TrackLab software (Noldus Information Technology, Wageningen, The Netherlands). Broilers were fitted with a battery powered Ubisense tag with a size of approximately 3.8 by 3.9 cm and a weight of ~25 g on their backs, using elastic bands around their wing base. Every 6.91 s, these tags sent out a signal. The sampling rate could be set higher, but a pilot study showed that the current sampling rate was best suitable [23]. With the sampling rate of 6.91 s, the batteries of the tags could last for at least seventeen consecutive days, working continuously. Before each trial, all batteries were tested and only full batteries were used. The room in which the broilers were housed was fitted with four Ubisense beacons, in a square-like structure above the pen at a height of ~2.25 m that could receive these signals. Using triangulation of the signal from the tags, based on the time of arrival of the signal (Time Difference of Arrival (TDoA)), the location of the tags could be determined. When the signal did not meet the error threshold of the Ubisense tracking system, which could for example occur when the tag was in a partially hidden position, such as in a corner or underneath the drinking line, and the signal could not be picked up by a number of beacons, the sample was seen as invalid by the system and was not recorded. The recorded locations of the tags were sent to the TrackLab software. The output of the TrackLab software that was used in this study was the total distance moved in meters per individual per tracking period.

### 2.4. Distance Validation Study

To validate the distances moved as recorded by the UWB system (referred to as ‘distance validation study’ throughout this paper), 24 male broilers of two weeks of age from two genetic crosses were used. These birds were taken from a larger group, selecting the lowest and highest body weights. From each cross, three heavyweight and three lightweight birds were selected for UWB tracking. The average weight (±SEM) of the light birds was 0.42 ± 0.01 kg, while the heavy birds weighed on average 0.63 ± 0.01 kg, as measured on 15 days of age. These twelve birds were fitted with an UWB tag (*n* = 6 per cross) and were colour-marked for identification purposes. During the study, one tag stopped working and was replaced with another tag, resulting in thirteen tags being used. The 24 broilers were housed in a pen with a size of approximately 6 m^2^ in an octagonal shape. The birds were tracked with the UWB system from day 15 to day 33 of life (*n* = 19 days), for approximately one hour each day, at different times. This one-hour sample per day was deemed sufficient as the main interest here was validating UWB recordings and not studying individual activity patterns over time. Video recordings were made from above, using a Zavio B6210 2MP (Zavio Inc., Hsinchu City, Taiwan) video camera, and were analysed using Kinovea video analysis software version 0.8.25. The length of one side of the octagonal pen (~1.15 m) was used for calibration of the distances for Kinovea. Manual corrections were applied where necessary, for example, when the bird was running, flapping its wings or was very close to other birds. To validate the UWB system, the distance moved according to the UWB system was compared to the distance moved as scored from video. One bird died, and consequently no data was available for day 29 to 33 for this animal, resulting in a total of 223 samples of recorded distances from both the UWB system and video tracking.

### 2.5. Activity Trends Study

To study individual levels of activity of broilers and assess group-level trends in activity over time (referred to as ‘activity trends study’ throughout this paper), 150 male broiler chickens from four crosses (A−D) were used, distributed over four consecutive trials (T1−T4; see Table 1). These birds were selected from a larger group of birds on day 13 or 14 of life, taking an equal sample of the lightest and heaviest birds for each cross and trial. During T1, T2 and T4, the broilers were housed in a pen with a size of around 6 m^2^ in an octagonal shape. This pen was divided into two equally sized compartments, one for each cross. In T3, the broilers were housed in a rectangular pen with a size of approximately 8 m^2^ in total. This pen was also divided into two equally sized compartments. In T3 and T4, extra birds that were not tracked were added to increase the density of birds to a level more comparable to commercial settings (generally up to 33 kg/m^2^ in the EU [24]; Table 1). The birds were tracked from 00:00 to 23:30 each day. The data from day 16 to day 32 of life (*n* = 17 days) was used for all trials, with exception of the birds in T4 that were tracked from day 18 onwards (*n* = 15 days) and the birds in T3, where no data was available for day 26 and 27 of life due to a system malfunction (*n* = 15 days). The tracking period was divided over five consecutive recording sessions of different durations. These sessions covered the periods between 00:00−03:30, 03:30−04:30, 04:30−07:00, 07:00−23:00 and 23:00−23:30. The half hour between 23:30 and 00:00 was not tracked in order to allow restarting the recordings each day. The UWB output was used to calculate the average distance moved per hour for each day and each bird. When over 10% of the samples were missing within a tracking session, which was true for approximately 1.4% of the tracking sessions, we considered that the tracking data was not complete for that day, and the average distance moved per hour was removed for this individual and day, resulting in a missing data point. The birds were weighed on day 13 or 14 and again on day 33, 34 or 35 of life (Table 1). Based on the start weight, individuals were categorised as lightweight or heavyweight within their respective trial and cross. Due to too much missing data (<75% of tracking days complete), death of birds (not related to the UWB tags) and mistakes in sexing, the final number of birds with tags was 32 in T1, and 35 in T2, T3 and T4 (Table 1). Overall, this resulted in a final sample size of 137 birds. For all trials, the mean weights of the broilers in the two weight categories are shown in Table 1.

### 2.6. Statistical Analysis

All statistics were performed using R version 3.5.0 (R Foundation for Statistical Computing, Vienna, Austria) [25]. For the distance validation study, the data was not normally distributed. A square root transformation normalised the data, but the results for untransformed and transformed data were very similar. Therefore, the untransformed data was used for analysis and is presented here. The correlation between the recorded distances from UWB and from video was studied using a repeated measures correlation (package rmcorr [26]), to correct for the repeated measures on the same tags, and a Pearson correlation. The Pearson correlation is presented here, as correcting for repeated measures only marginally affected the correlation. To study the relation between the recorded distances from the UWB system and video tracking in more detail, three groups (LD: low distance; MD: medium distance; HD: high distance) were created based on the distance moved on video, the boundaries for which were based on where the correlation line crossed the diagonal. 

For the activity trends study, the hourly activity data was not normally distributed, but a square root transformation did not improve the distribution. Therefore, untransformed data was used for the analysis and is presented here. To study how levels of activity are influenced by age and weight, a linear mixed-effects model was used (lme4 package [27]; lmerTest package [28]). The effects tested were day of tracking, cross, trial, start weight category, weight change, and the random effect of animal by day. A backward stepwise approach without interactions was used to test the effects. All effects were significant except for weight change. Using the resulting terms left, all possible two-way interactions were included, with exception of the interaction between cross and trial as not all crosses were present in each trial, and backward selection was again performed. After removal of the interaction between trial and weight, which was not excluded in the backward selection but was not a significant effect in the resulting model, the final model was
y_ijklm_ = μ + β(DT)_i_ + C_j_ + T_k_ + SW_l_ + (β(DT) × C)_ij_ + (β(DT) × T)_ik_ + (β(DT) × SW)_il_ + (1 + Day|ID_m_) + e_ijklm_(1)
where Y is the average distance moved per hour, μ is the overall mean, β(DT)_i_ is the ith day of tracking (i = 1–17), C_j_ is the jth cross (j = A–D), T_k_ is the kth trial (k = 1–4), SW_l_ is the lth start weight category (l = light or heavy), (β(DT) × C)_ij_ is the interaction between day of tracking and cross, (β(DT) × T)_ik_ is the interaction between day of tracking and trial, (β(DT) × SW)_il_ is the interaction between day of tracking and start weight category, (1 + Day|ID_m_) is the random effect of the mth animal by day and e_ijklm_ is the residual term. Visual inspection of the residuals indicated no obvious deviations from normality or homoscedasticity. *p*-values for the factors in the model were determined using the lmerTest package [28]. The R^2^ values of the model were determined using the MuMIn package [29]. Additional contrasts were calculated using the emmeans package [30]. Figures were made using the ggpubr [31], ggplot2 [32] and sjPlot [33] packages. The level of statistical significance was set at 0.05. In the text reported results are rounded to two decimals and activity levels are given in meters moved per hour.

## 3. Results

### 3.1. Distance Validation Study

The recorded distance moved on video was positively correlated with the recorded distance as calculated from the UWB tracking data (Pearson correlation, r = 0.71 (95% CI: 0.64–0.77), df = 221, *p* < 0.001). Figure 1 shows the resulting correlation. The diagonal is also indicated in this figure.

From Figure 1 it can be seen that the spread of the individual data points around the diagonal is not equal for the different distances moved on video. The correlation line crosses the diagonal at about 22 m moved on video (Figure 1). Using this crossing as a reference point, the following three distance groups, based on the distance recorded on video, were created; low distance with distances below 15 m, medium distance ranging from 15 to 30 m and high distance with distances over 30 m. For the distance groups and the overall data set, the mean and median proportional differences between video and UWB tracking, as well as the largest under- and overestimations, can be found in Table 2. Overall, the UWB system on average overestimated the distance moved by 10% compared to the distance recorded on video, while the largest underestimation was 63% less than the distance recorded on video. The largest overestimation that was observed was 345% more than the distance recorded on video. When comparing the three distance groups, the MD group had the best average estimation (3% overestimation), but also had the largest underestimation (63%). The LD group had the largest mean deviation (40% overestimation) and the largest overestimation (345% overestimation), while in the HD group the UWB system on average underestimated the distance moved (15% underestimation).

### 3.2. Activity Trends Study

The results of the linear mixed-effects model for the predicted average activity are shown in Table 3. Clear effects of day and trial on activity levels were observed, as well as an interaction between these factors. Furthermore, an effect of weight category was observed. The activity model explained 56.93% of the variance when only fixed factors were included, while it explained 85.50% of the variance when both the fixed effects and the random effect of ID by day were included.

The overall average activity in the study was found to be 18.65 m per hour. The predicted average activity decreased over time, regardless of differences between trials and crosses (Table 3; Figure 2). From Figure 2 it can be seen that in all trials the predicted average activity decreased over the tracking period. However, the activity was on average higher in T3 and T4, compared to T1 and T2. Furthermore, the degree of decrease in activity over time was higher in T1 and T3, compared to T2 and T4.

The predicted average activity per hour also differed between the two weight categories, as well as the degree of decrease in activity over time (Table 3; Figure 3). From Figure 3 it can be seen that heavyweight birds are on average less active and that the activity decreases faster in lightweight birds. Compared to heavyweight birds, lightweight birds are estimated to move on average 2.05 m per hour more (averaged over time and the levels of trial and cross: day = 9.18, estimate = 2.05, SE = 0.55, df = 137, t = 3.74, *p* = 0.0003; Figure 3).

## 4. Discussion

In this study, the distance recorded with the UWB system was moderately strong positively correlated with the distance recorded from video. Furthermore, we found that the UWB system detected a decrease in activity over the duration of the trials and detected that lightweight birds on average moved longer distances per hour than heavier birds. 

### 4.1. Distance Validation Study

The output of the UWB system was compared to video analysis using Kinovea to assess whether the UWB output was a reliable indicator of the actual distances moved. The moderately strong correlation between the output of the UWB system and video tracking using Kinovea indicates that the UWB system can provide reliable information on the distances moved by individual animals. Overall, the UWB system on average overestimates the ‘true’ distance moved by 10%. However, it does appear that the distance moved according to the UWB system is generally an overestimation of the actual distance moved as determined by video analysis when animals move less. When animals move more, the UWB system underestimates the distances moved (Figure 1). When looking at the different distance groups, this pattern is indeed observed. When broilers move low distances on video (<15 m), the distance recorded by the UWB system is on average an overestimation of 40%. However, when broilers move large distances on video (>30 m), the distance recorded by the UWB system is on average an underestimation of 15%. This could be due to the sampling rate that was implemented, i.e., 6.91 s in this study. With each sample that is received, there can be some noise; the triangulation-based location of the tag may deviate slightly from the actual location. Consequently, if an animal moves very little, this noise can make up a relatively large part of the total registered distance, which could explain the overestimation of distances moved by the UWB system when the actual distance moved is low. Alternatively, if an animal is very active, some of the movement of the animal might be missed by the system. For example, if an animal moves from position A to B and back to A within the specified sampling rate, the full distance covered is not registered. In this case, the sampling rate would result in the animal being located near position A two times in a row, while the movement to and from position B is (partly) missed. This could explain the underestimation by the UWB system of the distance moved by animals that covered larger distances on video. However, a pilot study [23], in which different sampling rates were compared for agreement between distances moved with the UWB system and on video, indicated that the sampling rate used in the current study was the best fit for our implementation. Concluding, the distances moved as indicated by the UWB system may not fully represent the exact distances moved by the animals, but the moderately high correlation indicates that the UWB system is suitable for making comparisons between individual animals. However, because the sampling rate can result in over- or underestimation, it is pivotal to compare all groups or animals using the same sampling rate.

### 4.2. Activity Trends Study

#### 4.2.1. Activity Levels over Time

This study showed a clear decrease in activity, measured here as average distance moved per hour, over time, as was expected from literature (e.g., the work by the authors of [19]). A possible mechanism that may underlie the decrease in activity that is observed is the increasing weight of the birds over time. Tickle et al. (2018) showed that level of activity and metabolic costs are inversely related over development. Heavy birds are relatively inactive, and this is hypothesized to be the result of locomotion being more energetically expensive in heavier birds [19]. Furthermore, it has been shown that there is a relationship between body weight and gait score, with heavier birds often having a higher gait score (i.e., a worse gait [4]). Weeks et al. (2000) compared gait scores 0, 1, 2 and 3 (ranging from no detectable abnormality to an obvious gait defect [6]) and found that the birds with higher gait scores spent less time walking [18]. Possibly, the increasing weight of the birds over time could result in a decrease in distance moved as a consequence of these higher gait scores, but this cannot be concluded from the current study and the relationship between activity, gait and weight at the individual level requires further investigation. However, the finding that the UWB tracking system is able to detect these changes in activity over time supports the notion that the UWB tracking system is suitable for monitoring individual activity of broilers.

In the current study, the average distance moved per hour over the full tracking period was found to be ~18.7 m. When the birds were ~4.5 weeks old, the average distance moved per hour was approximately 15.1 m. However, the literature indicates that there is much variability between studies in distances moved by broilers. In a broiler study by Lewis & Hurnik (1990) lower distances moved were recorded [34]. Their recorded distances varied between 8.1 and 10.0 m per hour, depending on the density at which the birds were housed (660–1320 cm^2^ per bird, respectively). In their study, the activity was recorded at about five to six weeks of age, and, given the declining trend in activity over time mentioned earlier, these results may match the findings in the current study. However, in a study by Sherlock et al. (2010) an average distance moved per hour of 46.1 m was found at six weeks of age [35]. This number was based on the number of gridlines crossed in their test and the average distance between them, and is higher than the distance found in the current study. 

One possible explanation for the discrepancy in the results is that the added weight of the UWB tags decreased the distance moved by the broilers in our study. When broilers are fitted with tags, about 25 g of weight is added. Possibly, this increase in weight reduced the activity of the broilers. In this study, no comparison between birds with and without tags was made, but Stadig et al. (2018) looked into the effects of fitting slow-growing broilers with UWB tags [36]. Using backpacks, 35-day-old birds were fitted with UWB tags, with a weight of ~36 g, equal to about 1.8% of the birds’ body weight at that time point. They compared birds with and without tags fitted and found an effect of the tag on walking behaviour in the first week after tagging (week six of life), with lower percentages of time spent walking for birds with tags, i.e., 5.8% for tagged birds and 8% for nontagged birds, respectively. This suggests that in the current study the tags may also have decreased the distance moved somewhat. However, there are several other possible explanations for the discrepancy in the results of the different studies. Activity in broilers has been noted to be influenced by numerous factors. For example, higher stocking densities are reported in literature to result in lower activity [34]. Another housing aspect that may influence broiler activity is the lighting. Blatchford et al. (2009) compared light intensities of 5, 50 and 200 lx, and found that broilers reared at 5 lx were less active during the day [37]. The distance between feeders and drinkers may also affect the recorded activity levels. Reiter (2004) compared two feeder-drinker distances—2 and 12 m—and found that the locomotor activity was higher when the distance was larger [7]. Overall, differences in the surroundings of broilers may affect the recorded distances moved and cause discrepancies between studies. 

#### 4.2.2. Differences in Activity between Weight Categories

This study showed that birds that are more lightweight at about two weeks of age are on average more active than heavier birds, and showed a faster decline in activity for lightweight birds compared to heavyweight birds. The difference in average activity is in agreement with the expectation from literature (e.g., [20]) and further supports the notion that the UWB system is suitable for tracking activity in broilers. As noted before, the decrease in activity that is seen over time is likely related to the increasing weight of broilers over time and this finding may also underlie the difference in activity that is observed between lightweight and heavyweight birds. Bokkers et al. (2007) studied high and low body weight groups (aiming at 90% and 50% of normal commercial conditions, respectively) from a fast-growing broiler hybrid in an operant runway test, where the broilers had to walk for a food reward [38]. Birds with a lower body weight were found to walk a larger distance in the test than heavier birds [38]. Reiter & Bessei (2001) studied locomotor activity of load-reduced and non-load-reduced birds. Load-reduced birds were fitted with a harness and suspension, to alleviate the weight load on the legs by 25 percent, while non-load-reduced birds received no weight alleviation [20]. It was found that the average distance travelled by broilers over four weeks, determined from video observations made two days per week, was higher in load-reduced birds compared to non-load-reduced birds [20]. Possibly, lower locomotor ability and pain related to a higher weight load on the legs underlie the lower activity of the non-load-reduced birds [20]. Rutten et al. (2002) also performed a study where the weight load of broilers on their legs was alleviated, by 50 percent, with a suspension mechanism [39]. The distance travelled by the broilers was found to be greater in the birds that received load reduction compared to birds that did not receive load reduction in the second week of the experiment, but not in the first week (day 6 to 12 of age [39]). Possibly, in the first weeks all birds are sufficiently lightweight to not experience any locomotor consequences of the load-bearing on the legs. In the current study, all birds were tracked from two-weeks-old onwards, when the weight may have already limited the locomotor activity of the birds, as is supported by the difference in average activity of lightweight and heavyweight birds that was found.

Furthermore, in the current study, the degree of decrease in activity over time was larger for lightweight birds, and consequently the difference in average activity between lightweight and heavyweight birds became less pronounced over time. However, when looking at the weight of the birds at the start and end of tracking (Table 1), the relative difference in the weight of the lightweight and heavyweight categories is smaller at the end of tracking compared to at the start. Lightweight birds are approximately 22% lighter at the start compared to the heavyweight birds, while the difference is approximately 10% at the end. Possibly, as the weights of the two categories approach each other over time, the difference in activity level also becomes less, as is reflected in the difference in slope of activity over time. However, more detailed recordings of body weight over time are required in future studies to confirm this hypothesis.

#### 4.2.3. Effects of Trial and Cross

Besides the overall effects over time and of different weight categories, effects of trial and cross in interaction with day were also observed. The four trials, over which this study was divided, differed in the average activity shown by the birds, as well as in the degree of decrease in activity over time. These differences between trials may have arisen from differences in the setup of these trials. First, the trials included different genetic crosses, which may have confounded cross and trial effects. It has been suggested in literature that broiler stocks may differ in activity due to genetic factors [8]. Therefore, a difference between crosses in degree of decrease in activity over time, as was observed here, is not unexpected. However, the distribution of crosses over the trials was skewed and a small number of broilers per cross was available, which hinders drawing conclusions from these findings. Another difference in the setup of the trials was the pen size and stocking density. The size of the pen was different in T3, compared to the other trials. In T3, the pen had a size of approximately 8 m^2^, while it had a size of approximately 6 m^2^ in the other trials. Furthermore, in T3 and T4, the stocking density was higher than in T1 and T2 (approximately 12 birds/m^2^ versus 6 birds/m^2^). In the current study the birds housed at higher densities were more active. By contrast, as discussed before, the opposite has been reported in literature [34]. In the current study, however, the stocking densities were relatively low compared to those reported in literature. The densities reported in the study by Lewis & Hurnik (1990) correspond to ~7.6–15.2 birds per square meter [34]. Blokhuis & Van der Haar (1990) studied densities between 2 and 20 birds, and found a decrease in percentage of birds walking with increasing density in week seven of life [40]. In our study, the number of birds per square meter varied between approximately 6 and 12. Therefore, the high density in the current study might actually be considered as moderate in comparison to literature and commercial situations. As a result, the restrictive element of a higher stocking density may not have applied here. Moreover, the higher stocking density in the current study may have resulted in increased activity, as a consequence of more disturbance by other birds. It has been noted that when broilers are housed at low densities, they preferentially remain close to feeders and drinkers, even though there is sufficient space available to move elsewhere [41]. However, when there is competition for space at the feeder and drinker, which might result at higher local densities, birds resting near the feeders and drinkers may be displaced, which may lead to higher distances moved. This might explain the higher activity found in T3 and T4 compared to T1 and T2. However, due to the differences between trials in the crosses studied, this cannot be conclusively determined with the current setup and requires more detailed investigation. Overall, however, it is evident that it is important to use a consistent study design when different trials are compared.

### 4.3. Activity as a Predictor

In this study, individual data on activity of broilers was used to study group-level patterns for validation of the UWB system. However, in future work it would be interesting to study individual activity patterns in more detail, to determine the predictive value of activity for growth or gait problems, for example. Furthermore, data on the first two weeks of life would be interesting to add, to allow assessment of whether distances moved throughout the production period can be predicted using data from the first few days of life. A study by Bizeray et al. (2000) has shown correlations between activity early in life and at later ages [8]. They found a positive correlation between activity at two to three days of age and at three weeks of age in fast-growing broilers. Such insights can be valuable for broiler breeding programmes. However, with the UWB tracking system used in this study, birds cannot be tracked in the first two weeks of life, due to the UWB tags being too large and heavy to wear at a young age. Possibly other tracking systems, such as passive RFID, have the potential to track activity of birds throughout life, but this requires further investigation.

## 5. Conclusions

In this study, it was shown that the implemented UWB system is suitable for tracking activity of individual broilers. There was a moderately strong positive correlation between the output of the UWB system and video tracking. Furthermore, the UWB system could detect reductions in activity over time and could detect that lightweight birds, as determined at about two weeks of life, are on average more active than heavier birds. However, it is important to keep all settings the same when comparing different birds and trials. The longitudinal information on differences in activity can potentially be used as a proxy for health, welfare and performance, but further research into individual patterns of activity is required.

## Figures and Tables

**Figure 1 animals-09-00580-f001:**
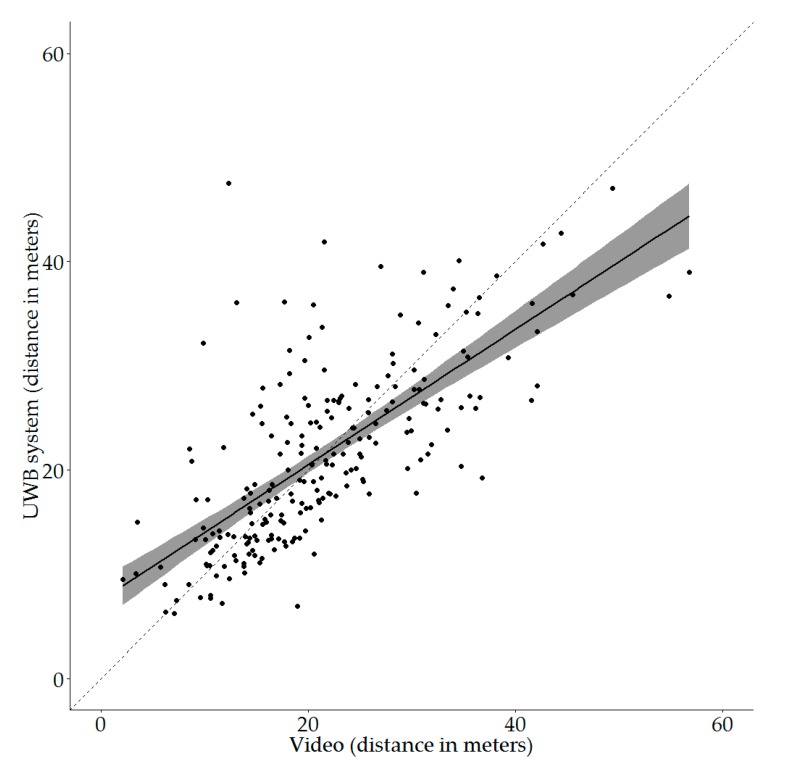
Plot of the correlation between the distances recorded from video observations using Kinovea and the distances recorded with the ultra-wideband (UWB) system. Dots represent individual data points. The solid black line shows the correlation coefficient, with the grey area representing the 95% confidence interval. The dashed line shows the diagonal where UWB and video distances would be exactly the same.

**Figure 2 animals-09-00580-f002:**
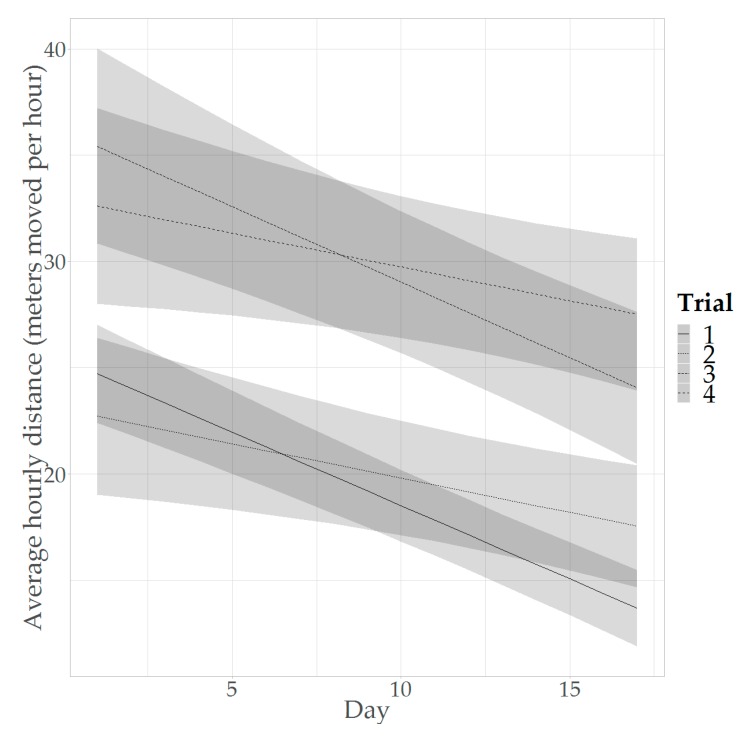
Predicted activity over the tracking period (day 1 to 17 of tracking, corresponding to day 16 to 32 of life) in the activity model, distinguishing between the different trials. Shaded areas represent 95% CIs.

**Figure 3 animals-09-00580-f003:**
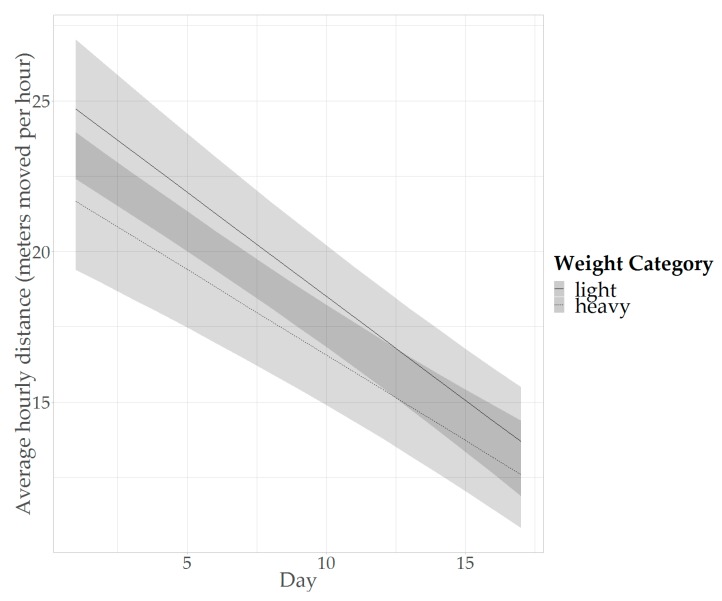
Predicted activity over the tracking period (day 1 to 17 of tracking, corresponding to day 16 to 32 of life) in the activity model, distinguishing between the different weight categories. Shaded areas represent 95% CIs.

**Table 1 animals-09-00580-t001:** Overview of sample sizes, densities, measurement days, weights (mean ± SEM; SW = start weight, EW = end weight) and average weight increase (mean ± SEM) for the activity trends study. The weights are separated for the two weight categories (L = light; H = heavy). Weights of individual birds were determined with five-gram precision and reported averages are rounded to five-grams. SEMs are rounded to round numbers.

Trial	Number of Tagged Birds (Start)	Birds without Tag Added	Density (Birds Per m^2^)	Number of Tagged Birds (end)	SW Day	EW Day	Weight Category	SW (g)	EW (g)	Average Weight Increase Per Day (g)
T1	36	no	~6	32	13	34	L (*n* = 16) H (*n* = 16)	420 ± 5 520 ± 4	2435 ± 43 2635 ± 60	95 ± 2 100 ± 3
T2	36	no	~6	35	13	33	L (*n* = 18) H (*n* = 17)	485 ± 7 595 ± 6	2450 ± 34 2680 ± 45	100 ± 2 105 ± 2
T3	40	yes	~12	35	14	35	L (*n* = 15) H (*n* = 20)	480 ± 12 630 ± 6	2500 ± 55 2715 ± 71	95 ± 2 100 ± 3
T4	38	yes	~12	35	13	35	L (*n* = 17) H (*n* = 18)	340 ± 17 460 ± 5	2155 ± 78 2520 ± 32	85 ± 3 95 ± 1
Total	150			137						

**Table 2 animals-09-00580-t002:** Under- and overestimations by the UWB system for the different distance groups and the complete data set. Proportional differences are calculated as ((UWB – Video)/Video).

Distance Group	*n*	Mean Proportional Difference	Median Proportional Difference	Largest Proportional Underestimation	Largest Proportional Overestimation
Low (<15 m)	59	0.40	0.10	−0.38	3.45
Medium (15−30 m)	122	0.03	−0.04	−0.63	1.03
High (>30 m)	42	−0.15	−0.16	−0.48	0.25
Total	223	0.10	−0.04	−0.63	3.45

**Table 3 animals-09-00580-t003:** Results of the linear mixed-effects model for the predicted average activity (meters moved per hour), including type III Analysis of Variance and estimates for the different factor levels.

Linear mixed-effects model					
**Random effects**					
**Factor**	**Variance**	**SD**	**Correlation**		
ID intercept	18.837	4.340	−0.72		
ID by Day	0.059	0.244			
Residual	5.707	2.389			
**Fixed effects**					
**Factor ^1^**	**F-value**	***p*** **-value**	**Estimate**	**SE**	***p*** **-value**
Intercept			25.413	1.235	<2 × 10^−16^
Day	337.322	<2.2 × 10^−16^	−0.690	0.074	4.73 × 10^−16^
Cross	2.313	0.079			
Cross B			−3.466	1.597	0.032
Cross C			−4.447	2.209	0.046
Cross D			−5.918	2.464	0.018
Trial	28.531	2.177 × 10^−14^			
Trial 2			−2.366	1.525	0.123
Trial 3			10.728	2.096	1.05 × 10^−6^
Trial 4			7.510	2.109	5.09 × 10^−4^
Weight category	16.665	7.545 × 10^−5^			
Heavyweight			−3.175	0.778	7.54 × 10^−5^
Day-Cross	3.112	0.029			
Day-Cross B			−0.023	0.096	0.810
Day-Cross C			−0.053	0.133	0.688
Day-Cross D			−0.255	0.149	0.089
Day-Trial	19.052	2.273 × 10^−10^			
Day-Trial 2			0.366	0.092	1.08 × 10^−4^
Day-Trial 3			−0.021	0.126	0.868
Day-Trial4			0.372	0.127	0.004
Day-Weight category	6.810	0.010			
Day-Weight category heavy			0.123	0.047	0.010

^1^ Interactions between factors are indicated with (-).

## Data Availability

The data that support the findings of this study are available from Cobb Europe but restrictions apply to the availability of these data, which were used under license for the current study, and so are not publicly available. Data are however available from the authors upon reasonable request and with permission of Cobb Europe.

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
