# Peer review of "Validation of an Ultra-Wideband Tracking System for Recording Individual Levels of Activity in Broilers"

_animals, 2019, doi:10.3390/ani9080580_

Round 1
Reviewer 1 Report
Line 115: italic for “ad libitum” Line 116: please add the reference for the lighting, temperature, vaccine program, such as: “Cobb Management Guideline”, etc..
Line 131: please double check the reference #22, is this a paper or thesis?
Line 146: please use “on 15 day of age” Line254: please change the format for Table 2 and 3 to meet the requirement of the journal
Line 352-355: Did the author conduct comparison study between the “normal birds” vs “with UWB tag birds” on the movement and activity, etc???
Line 397-406: Is there any possibility for these “smaller birds” showed compensatory effect on the later growth performance results??
Author Response
We thank you for your valuable input and for helping us to improve this manuscript. We have worked on the suggested changes and our response to each suggestion is noted below. All changes are also highlighted in the text of the manuscript.
Line 115: italic for “ad libitum”
We changed the font of “ad libitum” to italic.
Line 116: please add the reference for the lighting, temperature, vaccine program, such as: “Cobb Management Guideline”, etc..
We added a reference, referring to the Cobb Broiler Management Guide (line 117).
Line 131: please double check the reference #22, is this a paper or thesis?
This reference is a Master thesis. We now added this information to the reference and in the text mentioned this as a pilot study (line 130 and 322).
Line 146: please use “on 15 day of age”
We changed this to “on 15 days of age”.
Line 254: please change the format for Table 2 and 3 to meet the requirement of the journal
We adjusted the tables to meet the journal format for the tables.
Line 352-355: Did the author conduct comparison study between the “normal birds” vs “with UWB tag birds” on the movement and activity, etc.???
In this study, we did not make a comparison between birds with and without UWB tags, as all the birds in our study were tagged. However, because we cannot exclude an effect of this on the overall level of activity, we do mention that this could have affected our distance findings. To support this statement, we cite a study where the effect of tags was studied in detail. We now specifically state in the paper that we did not study this ourselves (line 361).
Line 397-406: Is there any possibility for these “smaller birds” showed compensatory effect on the later growth performance results??
If we understand correctly, the reviewer asks whether the smaller birds might have grown faster during the period between (approximately) day 14 and 33, to make up for their lower weight at the start of tracking. In this study, weight was only measured at the start and end of the study, and therefore we cannot conclusively state whether these birds grew faster to ‘compensate’ their lower weight at the start. In terms of absolute weight change between (approximately) day 14 and 33, it appears that the heavier birds grew more (as can be seen in Table 1), but whether this was a constant growth difference over time cannot be determined from our data.
Reviewer 2 Report
I thoroughly enjoyed reading this paper, it is good considered science and written very well.
I only have minor comments to consider.
Overall comments:
Is it possible to state if the UWB system requires batteries, and if so if these were required to be changed and how frequently. This may impact the practicalities of use in future research.
Methods
How were individuals identified on the video recordings?
Can you please further define your 'tracking session' in section 2.5 (e.g. duration & purpose of categorising sessions)
L136: Please consider defining, or rephrasing, 'partially covered position'
L145: Please add error to the average BW values stated
L180: How frequently were data points missing?
Author Response
We thank you for your valuable input and for helping us to improve this manuscript. We have worked on the suggested changes and our response to each suggestion is noted below. All changes are also highlighted in the text of the manuscript.
I thoroughly enjoyed reading this paper, it is good considered science and written very well.
Thank you for reading the manuscript.
I only have minor comments to consider.
Overall comments:
Is it possible to state if the UWB system requires batteries, and if so if these were required to be changed and how frequently. This may impact the practicalities of use in future research.
We have now added a part on the batteries and their replacement rate: in line 128 we state that the tags are battery-powered. In lines 131-133 we state “With the sampling rate of 6.91 seconds, the batteries of the tags could last for at least seventeen consecutive days, working continuously. Before each trial, all batteries were tested and only full batteries were used.”.
Methods
How were individuals identified on the video recordings?
The individuals were identified using colour-markings. We have added this information in the manuscript in line 150.
Can you please further define your 'tracking session' in section 2.5 (e.g. duration & purpose of categorising sessions)
We added additional information about the tracking sessions in lines 178-179. We now state “The tracking period was divided over five consecutive recording sessions of different durations. These sessions covered the periods between 00:00-03:30, 03:30-04:30, 04:30-07:00, 07:00-23:00 and 23:00-23:30.” These periods were initially created to match with the light and dark periods.
L136: Please consider defining, or rephrasing, 'partially covered position'
We changed this to “When the signal did not meet the error threshold of the Ubisense tracking system, which could for example occur when the tag was in a partially hidden position, such as in a corner or underneath the drinking line, and the signal could not be picked up by a number of beacons, the sample was seen as invalid by the system and was not recorded.”.
L145: Please add error to the average BW values stated
Thank you for noticing this, we accidentally left this out. We have added the SEM to the averages in lines 148-149.
L180: How frequently were data points missing?
We did not calculate an exact number of missing samples. However, we excluded recording sessions (by animal) when more than 10% of the maximum number of samples were missing. The maximum number of samples was determined per recording session for other individuals. This resulted in approximately 1.4% of the recording sessions being left out. This is now stated in lines 181-184: “When over 10% of the samples were missing within a tracking session, which was true for approximately 1.4% of the tracking sessions, we considered that the tracking data was not complete for that day, and the average distance moved per hour was removed for this individual and day, resulting in a missing data point.”.